# Insomnia, Cognitive Impairment, or a Combination of Both, Alter Lipid Metabolism Due to Changes in Acylcarnitine Concentration in Older Persons

**DOI:** 10.3390/metabo15060417

**Published:** 2025-06-19

**Authors:** Selma Karime Castillo-Vazquez, Berenice Palacios-González, Marcela Vela-Amieva, Isabel Ibarra-González, Ricardo Morales, Paola García-delaTorre, Sergio Sánchez-García, Carmen García-Peña, Ricardo Reyes-Chilpa, Raúl Hernán Medina-Campos, Jessica Hernández-Pineda, Juan Carlos Gomez-Verjan, Nadia Alejandra Rivero-Segura

**Affiliations:** 1Dirección de Investigación, Instituto Nacional de Geriatría, Ciudad de México 10200, Mexico; selmakarime@gmail.com (S.K.C.-V.); raulmdc81@gmail.com (R.H.M.-C.); jverjan@inger.gob.mx (J.C.G.-V.); 2Posgrado en Ciencias Biológicas, Universidad Nacional Autónoma de México, Ciudad Universitaria, Ciudad de México 04510, Mexico; 3Dirección de Investigación, Instituto Nacional de Medicina Genómica (INMEGEN), Ciudad de México 14610, Mexico; bpalacios@inmegen.gob.mx; 4Laboratorio de Envejecimiento Saludable del INMEGEN en el Centro de Investigación sobre Envejecimiento (CIE-CINVESTAV Sede Sur), Ciudad de México 14330, Mexico; 5Laboratorio de Errores Innatos del Metabolismo, Instituto Nacional de Pediatría (INP), Ciudad de México 04530, Mexico; mvelaa@pediatria.gob.mx (M.V.-A.); lrmg66@gmail.com (R.M.); 6Instituto de Investigaciones Biomédicas, IIB-UNAM, Ciudad Universitaria, Ciudad de México 04510, Mexico; icig@iibiomedicas.unam.mx; 7Unidad de Investigación Epidemiológica y en Servicios de Salud, Área Envejecimiento, Centro Médico Nacional Siglo XXI, Instituto Mexicano del Seguro Social, Mexico City 06720, Mexico; pgarciatorre@gmail.com (P.G.-d.); sergio.sanchezga@imss.gob.mx (S.S.-G.); 8Dirección General, Instituto Nacional de Geriatría, Ciudad de México 10200, Mexico; mcgarciapena@gmail.com; 9Instituto de Química, Universidad Nacional Autónoma de México, Ciudad Universitaria, Ciudad de México 04510, Mexico; chilpa@unam.mx; 10Departamento de Infectología e Inmunología, Instituto Nacional de Perinatología, Isidro Espinosa de los Reyes, INPerIER, SSA, Ciudad de México 11000, Mexico; elocho.jhp@gmail.com

**Keywords:** insomnia, cognitive impairment, aging, acylcarnitines, lipid metabolism

## Abstract

Background/Objectives: Insomnia has been widely associated with cognitive impairment (CI). However, the relationship between the two entities (insomnia and CI) is poorly understood. In this context, adults with insomnia show metabolic changes, including alterations in the catabolism of branched-chain amino acids, glycerophospholipids, and glutathione and glutamate biosynthesis. Nevertheless, aging itself induces metabolic changes that may be amplified by chronic diseases that compromise the health of the elderly. Therefore, in the present study we aim to characterise metabolomic profiles of insomnia and CI alone in order to address a significant gap in current research regarding the pathways through which insomnia may lead to CI in older persons. Methods: In this study we perform a targeted metabolomics analysis (UPLC-MS) on 80 serum samples from the Cohort of Obesity, Sarcopenia, and Frailty of Older Mexican Adults (COSFOMA); these samples were classified into four groups (control, insomnia, CI, and insomnia + CI). Results: Our results show that insomnia increases the concentration of acylcarnitines (C10, C8, C14, C12:1, C18:1 and C18) as compared to CI group, while older persons with CI show a decrease the concentration of the acylcarnitines C16, C10 and C8. Finally, individuals with both conditions (insomnia + CI) show that only the concentration of the acylcarnitine C16 decreases compared to controls. Conclusions: Taken together, our results shed light on the relevance of insomnia on lipid metabolism in older persons.

## 1. Introduction

Aging and metabolism are undoubtedly related to each other; in fact, several authors have reported that metabolism slows down with age, leading to weight gain and other metabolic issues, including defects in metabolic pathways such as gluconeogenesis, lipogenesis, glycogen synthesis, and glucose uptake [1,2]. Moreover, it has been suggested that age-related metabolic changes are potentiated by unhealthy lifestyles, including short physical activities, poor diets, and poor quality of sleep [3].

Insomnia, characterised by difficulty initiating and maintaining sleep, early morning awakenings with inability to return to sleep, and daytime impairments including fatigue, decreased energy, and mood disturbances [4,5], is the most prevalent sleep disorder in older persons [6], and over the past 20 years, it has been widely associated with increased risk of cognitive decline and dementia [7], suggesting a bidirectional relation among both entities. However, there are still gaps in the literature that, if filled, may contribute to improving the quality of life of the elderly and to therapeutic interventions to prevent and reduce cognitive decline and the risk of dementia.

In this context, metabolomics (a method of analysing metabolites in biological samples for identifying biomarkers for predicting, diagnosing, and tracking diseases) [8,9] has been widely used to explore the metabolome in both insomnia and cognitive impairment, separately. For instance, it has been reported that adults with insomnia (25–50 y.o.) have alterations in the catabolism of branched-chain amino acids (BCAAs), including 2-oxoisocaproate and 3-methyl-2-oxovalerate [10]. Furthermore, other studies report that alterations in the microbiota of young individuals with insomnia induce alterations in metabolic pathways related to glycerophospholipids, glutathione, nitrogen, aminoacyl-tRNA, alanine, aspartate, and glutamate biosynthesis [11,12]. However, none of these studies were performed in older persons. There is only one study performed in older persons (75.90 ± 4.01 y.o.) with insomnia or short sleep duration (≤6H) that sought to understand the effect of insomnia on the microbiota metabolome, suggesting that insomnia increases levels of metabolites related to microbial metabolism, including benzophenone, pyrogallol, 5-aminopental, butyl acrylate, kojic acid, and deoxycholic acid [13].

On the other hand, metabolomic studies performed in individuals with mild cognitive impairment (MCI), an early stage of memory loss, or other cognitive ability loss [14] report alterations in the metabolism of sphingolipids, choline and purines [15], BCAA catabolism, glycolysis, and gluconeogenesis [16]; other studies demonstrated that individuals with MCI and Alzheimer’s disease (AD) exhibit change in the biosynthesis of primary bile acids, fatty acids, and unsaturated fatty acids [17]. However, none of the studies mentioned above aim to understand the relationship between insomnia and cognitive decline. Therefore, in the present study we aim to characterise metabolomic profiles of insomnia and CI alone or in combination to address a significant gap in the current research regarding the pathways through which insomnia may lead to CI in older persons. By doing so, we aim to address a significant gap in current research regarding the potential pathways through which insomnia may lead to CI in this population.

## 2. Materials and Methods

### 2.1. Cohort Description

We used 80 serum samples from the fourth wave (2018) of the Cohort of Obesity, Sarcopenia, and Frailty of Older Mexican Adults (COSFOMA). This prospective population-based study is of older persons ≥60 who are beneficiaries of the Mexican Social Security Institute (IMSS, by its Spanish acronym) and live in Mexico City. COSFOMA includes sociodemographic and clinical information [18].

### 2.2. Sample Collection

By forearm puncture, five hundred twenty-three samples were collected in 2018 at the Centro Médico Siglo XXI (IMSS institution) facilities. Blood samples were collected in the morning after a 12-h fast in red tubes from the BD Vacutainer. They were stored at −80 °C until use. We used 80 serum samples for this study, randomly selected according to the Insomnia Measure and Cognition Status Evaluation. Such samples were divided into four groups:

Control (no insomnia nor cognitive impairment) (*n* = 22), insomnia (*n* = 21), cognitive impairment (CI, *n* = 23), and both conditions (insomnia + CI, *n* = 14).

### 2.3. Insomnia Evaluation Symptoms

The Athens Insomnia Scale (AIS) is a wide-use self-report measure to evaluate insomnia symptoms and severity according to the International Classification of Diseases criteria version 10 (ICD-10) [19,20]. This test has been validated in Mexico and consists of eight self-rating items. The first four items measure sleep problems quantitatively, the fifth item assesses sleep quality, and the remaining three evaluate the impact of insomnia during the day. The scores range from 0 to 24, with a score of ≥6 indicating the presence of insomnia [21]. The AIS Cronbach alpha value in this study was 0.90.

### 2.4. Cognition Status Evaluation

The cognitive status was determined using the Mini-Mental State Examination (MMSE) test, a scale comprising 30 items that assesses various cognitive domains, validated for the Mexican population. The cut-off point was adjusted according to the level of schooling, with a value ≤23 indicating potential CI [22,23]. Here, we used the MMSE to identify individuals with possible CI.

### 2.5. Targeted Metabolomic Determinations

A targeted metabolomic analysis was conducted using liquid chromatography-electrospray ionisation positive-tandem mass spectrometry (LC-ESI-MS/MS), with a Quattro micro-API tandem MS in multiple reaction monitoring (MRM) mode and a commercial kit (NeoBase Non-derivatized MSMS Kit; PerkinElmer, Waltham, MA, USA) at the Laboratorio de Errores Innatos del Metabolismo from the Instituto Nacional de Pediatria. Briefly, the NeoBase Non-derivatized MSMS Kit is intended for the semi-quantitative measurement and evaluation of 40 analytes, including amino acids, succinylacetone, free carnitine, and acylcarnitine concentrations.

The analysis yielded the identification of 11 amino acids (AA), including ornithine (ORN), tyrosine (TYR), L-phenylalanine (PHE), alanine (ALA), glycine (GLY), citrulline (CIT), arginine (ARG), leucine (LEU), L-methionine (MET), Valine (VAL), and Proline (PRO), as well as succinylacetone (SA) and 28 acylcarnitine, and including free carnitine (C0), acetylcarnitine (C2), propionyl carnitine (C3), methylglutarylcarnitine (C6DC), isobutyric-L-carnitine (C4), isovalerylcarnitine (C5), tiglylcarnitine (C5:1), hexanoylcarnitine (C6), octanoyl carnitine (C8), acylcarnitine C8:1, L-palmitoylcarnitine (C16), acylcarnitine C16:1, acylcarnitine C16: 1OH, decanoylcarnitine (C10), acylcarnitine C10:1, acylcarnitine C10:2, dodecanoylcarnitine (C12), acylcarnitine C12:1, tetra-decanoylcarnitine (C14), acylcarnitine C14:1, acylcarnitine C14:2, acylcarnitine C14:OH, stere aroyl carnitine (C18), acylcarnitine C18:1, acylcarnitine C18:1OH, acylcarnitine C18:2, and acylcarnitine C18OH. The kit included all the reagents for the measurement, such as calibration standards, quality controls, and the mobile phase.

### 2.6. Sample Processing

The serum samples were thawed and vortex-homogenized; 20 μL of the samples was subsequently added to filter paper cards (Whatman 903, Dassel, Germany) and allowed to dry at room temperature. Automatic punching devices were used to create 3 mm circles from each dried serum sample, which were then transferred to a 96-well plate. Metabolite extraction was conducted according to the instructions provided in the NeoGram AAAC Spectrometry kit (Perkin Elmer, MA, USA). A volume of 190 μL of the working extraction solution, comprising a mixture of the corresponding stable isotope-labelled internal standards, was added to each well. The plate was then covered with aluminium foil and shaken at 650 rpm for 30 min at 30 °C. Subsequently, the plate was transferred to a 2777 C Waters auto-sampler (Waters Corp., Milford, MA, USA) for analysis.

A volume of 30 μL of the sample extract from each individual was injected into an HPLC pump (Waters 1525μ, Waters Corp., Milford, MA, USA). The samples were introduced without the previous chromatographic step at a 1.5 mL/min flow rate over a 1.5-min analysis period. A blank sample, composed of the extraction solution with internal standards for control purposes, was also included. Analytical controls and quality control samples were added. Analytes were ionised by electrospray ionisation using nitrogen as the curtain gas. The collision gas used between the two mass spectrometers was argon. For data acquisition, MRM mode was used. The signal ion intensity of metabolites was corrected to the corresponding internal standards, followed by calculating the concentration using MassLyns^®^ 4.0 software (Perkin Elmer-WallacTM Oy, Turku, Finland).

### 2.7. Assessment of the Acylcarnitine (AC)/L-Carnitine (LC) Ratio

The AC/LC ratio is considered a representative measure of disturbed mitochondrial metabolism. A cut-off value of >0.4 is considered abnormal and indicative of disturbed mitochondrial metabolism [24]. Data from the acylcarnitines C16OH/C16, C16/C2 and C18/C2 ratios were analysed by ANOVA followed by the Tukey test.

### 2.8. Statistical Analysis

The demographic and clinical description data were obtained from the study population. These data included the following: y.o in mean ± SD, the percentage for sex, education, living arrangement, medication, antidepressant consumption, anxiety, current smoking, current alcohol consumption, and multimorbidity. The multimorbidity variable was constructed based on considering more than two chronic diseases.

The quantified data underwent row-normalization by sum and data scaling via range scaling. The normalisation and the statistical analyses were conducted using the MetaboAnalist 6.0 (freeware available at http://www.metaboanalyst.ca, accessed on 29 April 2025) [25]. Partial least-squares discriminant analysis (PLS-DA) was employed to discern the distinctions between the samples, followed by a 2000-fold permutation test. The differential variables between the groups were determined using the scatter plot. The metabolites were then ranked according to their importance using the variable importance in the projection (VIP) plot, with a cutoff value of VIP > 1.0. Furthermore, significant differences in metabolite concentrations between groups were identified through ANOVA and using Tukey’s Honestly Significant Difference (Tukey’s HSD) post hoc tests. Furthermore, a pairwise analysis of groups was conducted to identify significant metabolites concerning the following: CI vs. control, insomnia vs. control, insomnia + CI vs. control, CI vs. insomnia, CI vs. insomnia + CI, and insomnia vs. insomnia + CI. The quantified data of CI vs. control underwent data transformation via log transformation and data scaling via mean centring. Insomnia + CI vs. control data underwent normalisation by sum, data transformation via square root transformation, and data scaling via auto-scaling. The quantified data of CI vs. insomnia underwent normalisation by sum, data transformation via cube root transformation, and data scaling via auto-scaling. PLS-DA and VIP analyses were performed for pairs of the groups with data previously normalised, followed by a 2000-fold permutation test. Furthermore, *t*-tests were performed for each pair of groups, except for insomnia + CI vs. control, for which a Wilcoxon rank-sum test was employed.

### 2.9. Ethical Statement

The COSFOMA protocol was approved by the IMSS National Committee of Research (the National Committee for Scientific Research and the Ethics Committee on Health Research) (Registration No. 2012-785-067). All participants and their legal guardians were informed of the research procedures and signed a consent letter before participating. For illiterate participants, written informed consent was taken from their legal guardians. Additionally, the use of samples for this study was approved on 14 March 2024 by the Comité Nacional de Investigación Científica-IMSS with the number R-2023-785-027. All methods employed in the study followed the Declaration of Helsinki and guidelines from the Ley General de Salud of Mexico.

## 3. Results

This study included 80 participants. Table 1 depicts the sociodemographic and clinical variables of this population. According to this table, our data are fairly homogeneous across the different groups. The results for the variables that could potentially impact our metabolomic analysis (BMI, fat mass, or weight) do not show any statistically significant differences (Appendix A).

### 3.1. CI Induces Alterations of C16, C10, and C8 in Older Persons

First, we perform a global metabolomic analysis comparing all the studied groups (control, insomnia, CI, and insomnia + CI) between them. Accordingly, our results show that all experimental groups share metabolites, and only six (ALA, ARG, acylcarnitine C5OH-C4DC, acylcarnitine C16, acylcarnitine C10:2, and acylcarnitine C5DC-C6OH) are different. However, the ANOVA and statistical parameters (Accuracy = 0.28301, R2 = 0.10064, and Q2 = −0.16206) indicate the model’s low predictability (Appendix A). Thus, we performed the metabolomic analysis by comparing the metabolomic profiles and organizing them into pairs. The PLS-DA shows a 50.4% of variance between CI vs. control (Figure 1a). Such variance may be explained by the VIP score obtained for the acylcarnitines C8, C16, C10, C14, and C5OH-C4DC (Figure 1b); however, when we compared the concentration of these metabolites, we found statistical differences only in acylcarnitines C16, C10, and C8, which increased significantly in the CI group as compared to the control group (Figure 1c).

### 3.2. The Metabolomic Profile of Insomnia Is Opposite to the CI Profile

Subsequently, we compared the metabolomic profiles of the insomnia and CI groups. The results from this analysis revealed that despite both groups sharing some metabolites (Figure 2a), fifteen metabolites are different between them (Figure 2b); among these metabolites highlight that their concentrations are opposite between groups. For instance, the concentration of acylcarnitines C10, C8, C14, C12:1, C18:1, C18, C14:2, C18:2, C16, C14:1, C2, and C5 increase in the insomnia group, while they decrease in the CI group. The same occurs for PHE and TYR (increase in CI and decrease in insomnia). Nevertheless, statistical analysis from Figure 2c shows that only acylcarnitines C10, C8, C14, C12:1, C18:1 and C18 are significant for the metabolomic profile.

### 3.3. Only C16 Increases in Older Persons with Both Conditions (Insomnia + CI)

Another interesting result from the metabolomic analysis reveals that when we compare metabolomic profiles from the group insomnia + CI with the control group, sixteen metabolites (acylcarnitines C2, C6, C14:2, GLY, ARG, C5OH-C4DC, PHE, VAL, acylcarnitines C8, C18, C10, C10:2, ALA, C4DC-C6OH, and C16) may constitute the metabolome of older persons with insomnia + CI (Figure 3a,b). Nevertheless, the statistical analysis reveals that only C16 significantly differs among these groups (Figure 3c).

### 3.4. Impairments in C16OH/C16 Ratio

Once we observed that both insomnia and CI, alone or in combination, impairs lipid metabolism, we aimed to explore if such results were mediated by mitochondrial impairment. Thus, we analysed the following ratios, C16OH/C16, C16/C2, and C18/C2 (Table 2), as an indirect measure of mitochondrial activity. Interestingly, the results demonstrate non-significant differences among groups (ANOVA followed Tukey test, *p*-value = 0.1550) suggesting that mitochondrial function may not be compromised. However, the C16OH/C16 ratio is increased as compared to the cut-off (0.1–0.4) suggested by [24,26,27].

## 4. Discussion

Multiple researchers suggest that metabolic processes are significantly altered during aging, coinciding with bodily transformations such as increased fat tissue and muscle loss [1,28]. In addition, such changes have been linked to the development of age-related diseases, including cardiovascular, metabolic, and neurodegenerative diseases [29,30]. However, current evidence suggests that lifestyle behaviours, including diet, physical activity, smoking, alcohol consumption, and sleep patterns, may contribute to the modulation of many underlying metabolic changes [3]. Specifically, our understanding of both sleep-related metabolic alterations in older individuals and the metabolic mechanisms linking insomnia and cognitive impairment (CI) remains limited. This study employs a targeted metabolomics approach to set bases for understanding the interconnected relationship between insomnia and CI. As the main limitations of this study, we highlight the size of the cohort and the bias that target metabolomics represent, since the NeoBase Non-derivatized MSMS Kit only measures 40 metabolites.

In this context, our main results demonstrate that insomnia impairs lipid metabolism by altering six acylcarnitines (C10, C8, C14, C12:1, C18:1 and C18). However, according to the results from both the PLS-DA models and the VIP analysis, a few AA such as PHE, TYR, ALA, VAL, ARG, and GLY may contribute to explain the variance between groups. The PLS-DA is a multivariate discriminant analysis used in metabolomics for predictive and descriptive modelling, allowing to differentiate groups and identify variables of major importance for separation using the VIP score >1.0 [31,32]. However, in metabolomics, a high VIP score indicates that a metabolite is important for separating groups, it doesn’t necessarily mean that its concentration differs statistically between the groups [33]. These results highlight the relevance of validating our results in larger cohorts.

On the other hand, acylcarnitines are essential for transporting fatty acids into mitochondria and have a predominant role in β-oxidation, contributing to energy metabolism. Interestingly, it has been reported that decreased mitochondrial activity (an antagonistic hallmark of aging) [34] results in increased levels of acylcarnitines, which promote proinflammatory signalling and are associated with age-related diseases such as neurodegenerative ones [35]. In this sense, C10 (decanoylcarnitine) concentrations decrease in CI while insomnia increases C10 concentrations. This result aligns with previous reports, which show that C10 increases in young adults (23 ± 5 y.o.) with acute sleep deprivation, leading to potential alterations in β-oxidation [36]. Regarding CI, C10 concentrations are pretty controversial [37], many authors report that C10 increases progressively from MCI to AD [38,39,40], and others report that C10 concentrations decrease in subjective memory complaints, MCI, and AD [41,42]. Our results are consistent with those reporting decreased C10 concentration in individuals with CI. Still, it is essential to note that since this is an exploratory analysis, we did not classify our CI group by MMSE score, and further association studies between acylcarnitine concentrations and variables that may influence them can be performed to shed light on new potential biomarkers for different stages of CI in older persons.

Our data show that C8 (octanoyl carnitine) decreases in CI while increasing in insomnia. Our findings are consistent with [41], who observed decreased C8 concentration in serum samples from individuals with MCI. In addition, reduced levels of C8 are significantly correlated with cognitive performances and lower MMSE scores in AD [43]. Conversely, the increase in C8 in older persons with insomnia in comparison to CI is limited in the literature. However, increased levels of C8 have been reported in prediabetes [44], type 2 diabetes (T2D) [45], and cardiovascular disease [46]. This is interesting, considering that previous studies mentioned that the metabolome of adults with insomnia suggests the presence of a prediabetic phenotype [10].

Elevated levels of C12:1 (dodecenoylcarnitine) have been associated with obesity, heart and muscle diseases, including hypertrophic cardiomyopathy, dilated cardiomyopathy, congestive heart failure, muscle weakness [47,48], and mitochondrial dysfunction in T2D. However, to date, there have been no reports of an association between insomnia and elevated levels of C12:1 in older persons. Regarding CI, our data are consistent with [43], who report a significant decrease in C12:1 concentration in both MCI and AD individuals, as well as the direct association between MMSE score and C12:1 levels, suggesting that C12:1 may be an interesting metabolite to intervene opportunity and prevent AD or CI.

Another acylcarnitine affected by insomnia is C14 (myristoylcarnitine). This long-chain acylcarnitine is generated as the product of L-carnitine esterification with long fatty acids obtained from the diet or by lipogenesis. The concentration of C14 varies according to food intake, meaning that in fasting, C14 increases, and in the fed state, it decreases. However, recent evidence summarised in [48] suggests that T2D [45], cardiovascular mortality in chronic kidney disease [49], and sleep deprivation [50] also increase C14 levels. Our data align with Cho et al. and may contribute to explaining the broad evidence about the association between insomnia or sleep deprivation and a high risk of cardiovascular diseases [51]. In contrast to our results, many authors report that C14 concentrations increase in MCI, SMC, and AD [37]. The divergence between our data and those of other previously reported studies may be due to several factors, such as diet, cognition state, or multimorbidity, and the effect of these variables may be addressed in further research.

C16 (palmitoyl-L-carnitine) is a long-chain carnitine identified as a key metabolite involved in the mitochondrial metabolism of palmitic acid. This metabolite has been associated with AD. Moreover, murine models (2–21 months old) demonstrated that both serum levels of palmitoyl-L-carnitine and tau phosphorylation increase with age. Tau phosphorylation is a pathological AD biomarker [52]. Conversely, a significant correlation has been observed between plasma C16 concentrations and cognitive performance, measured by the MMSE, in individuals with AD [43].

Additionally, it has been reported that serum C16 concentrations increase in individuals with MCI, while they decrease significantly in individuals with AD [41]. This study shows decreased C16 levels in the insomnia + CI group. This observation is noteworthy because most study participants were assessed with MCI. In metabolomic studies with adults with significant depression, with symptoms of insomnia and reduced appetite, a decrease in C16 was observed [53].

Our data show that acylcarnitines C18:1 and C18 decrease in CI while increasing in Insomnia; such results are consistent with other studies which have also observed a decrease in C18 concentration in serum samples from individuals with MCI and AD as compared to controls [41]. In addition, reduced levels of C18:1 have also been significantly correlated with cognitive performances and lower MMSE scores in AD [43]. In contrast, our results show that C18:1 increased in older persons with insomnia. This finding suggests a potential relation with [54], indicating that a single night of sleep restriction leads to a significant increase in C18:1 (>22% of concentration).

Finally, since acylcarnitines are related to mitochondrial metabolism, we analysed the ratios between C16OH/C16, C16/C2, and C18/C2 to evaluate if changes in C16, C18, and C18:1 acylcarnitines concentrations were related to impairments in mitochondrial metabolism. Interestingly, our data demonstrate that only the C16OH/C16 ratio was higher than 0.4; according to the literature an AC/LC > 0.4 means a potential mitochondrial dysfunction [24]. In this sense, C16OH/C16 indicates alterations of the mitochondrial trifunctional protein (MTP), a multienzyme complex implicated in the β-oxidation of long-chain fatty acids [26,55]. The impairment of MTP, leads to the accumulation of long-chain fatty acids that promotes lipotoxicity and oxidative stress [56], membrane disruption [57,58], and concomitantly energy deficits [59]. However, it is important to note that few reports in the literature perform the C16OH/C16 ratio among adults [27]. Hence, this is the first study to our knowledge that reports the mitochondrial impairment due to the increase of C16OH/C16 ratio. Such an increase may be related to the inherent mitochondrial dysfunction characterized in aging [34], and more studies are required to explore subtle shifts in this ratio or its relationship with other metabolic markers in longitudinal aging studies.

## 5. Conclusions

In conclusion, although changes in acylcarnitines have been previously described in sleep-deprived individuals [60], it is essential to note that this condition is not the same as insomnia. Therefore, in the present study, we report for the first time to our knowledge that insomnia preferentially promotes alterations in lipid metabolism and suggest that the concentration of acylcarnitines, C10, C8, C14, C12:1, C18:1 and C18, may be potential targets for improving health in older persons with insomnia. This study contributes to set the basis around the understanding of the biochemical changes induced by both insomnia and CI, separately or in combination, and contributes for the design of further studies that help to understand the bidirectional relationship between insomnia and CI in the Latin American population, which is underrepresented in other studies on sleep disorders.

## Figures and Tables

**Figure 1 metabolites-15-00417-f001:**
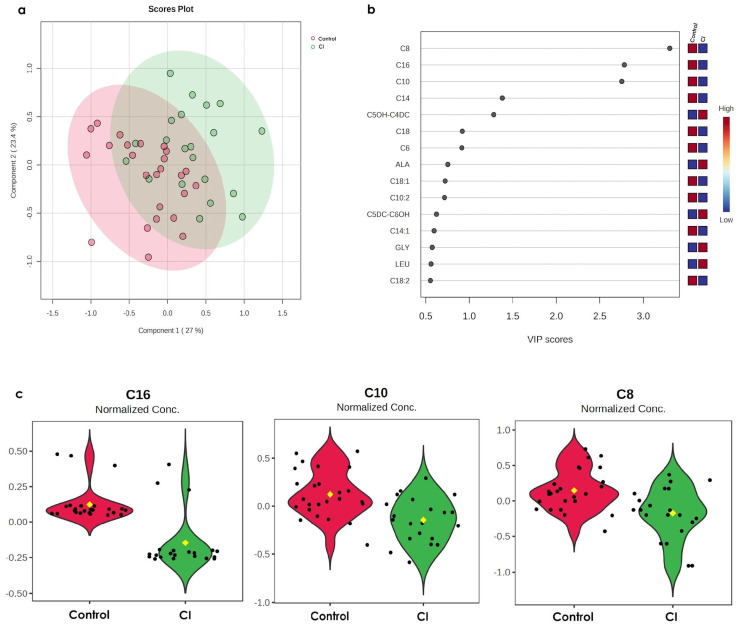
Both multivariate and bivariate analyses of metabolomic profiles reveal that acylacylcarnitines C16, C10, and C8 are significantly decreased in older persons with only CI. (**a**) PLS-DA analysis of relative concentration occurring in serum metabolome in the control group (red dots represent the metabolome for each individual in the group) vs. the CI group (green dots represent the metabolome for each individual in the group). The proportion of variance corresponds to Component 1: 27% and Component 2: 23.4%, Accuracy: 0.74444, R2: 0.33103, Q2: 0.2166, and the permutation *p*-value = 5 × 10^−4^ (**b**) VIP analysis represents the relative contribution of metabolites to the variance between groups. The colour scale (on the right side) represents metabolite concentration ranging from low (blue) to high (red). The VIP score cut-off has been adjusted to 1.0. Each dot (black) represent the VIP score calculated for the differential metabolites in both groups. (**c**) Violin plots represent the *t*-test of the normalised concentrations of the significant metabolites (C16 = 0.0232 ± 0.0018 mM (control) vs. C16 = 0.0152 ± 0.002 mM (CI), *p*-value = 4.45 × 10^−4^, FDR (False Discovery Rate) = 1.82 × 10^−5^; C10 = 0.236 ± 0.02 mM (control) vs. C10 = 0.1454 ± 0.02 mM (CI), *p*-value = 4.45 × 10^−4^, FDR = 0.0082391; C8 = 0.1408 ± 0.02 mM (control) vs. C8 = 0.0858 ± 0.01 mM (CI), *p*-value = 0.0016949, FDR = 0.020904). Dots (black) represent the metabolite concentration in each participant in the group, and the yellow dot in the centre represent the median of the corresponding metabolite concentration.

**Figure 2 metabolites-15-00417-f002:**
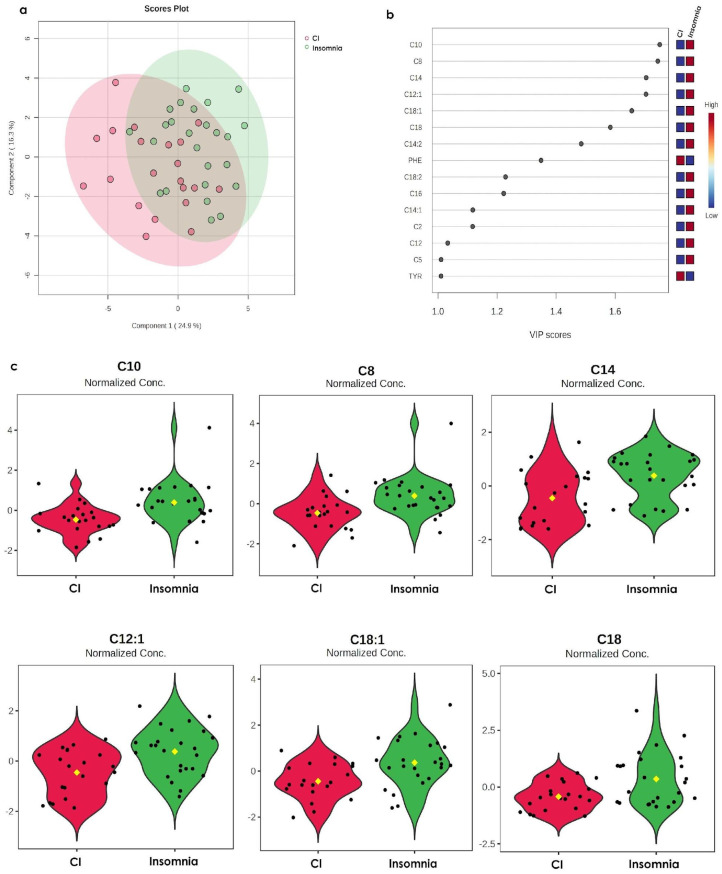
Comparison of the metabolomic profiles from the insomnia group and the CI group. (**a**) PLS-DA analysis of the relative concentration in the serum metabolome in the CI group (red dots represent the metabolome for each individual in the group) vs. the insomnia (green dots represent the metabolome for each individual in the group) group. The proportion of variance corresponds to Component 1: 24.9% and Component 2: 16.3%, Accuracy: 0.67333, R2: 0.28755, Q2: 0.14853, and the permutation *p*-value = 0.79. (**b**) VIP analysis represents the relative contribution of metabolites to the variance between groups. The colour scale (on the right side) represents metabolite concentration ranging from low (blue) to high (red). The VIP score cut-off has been adjusted to 1.0. Each dot (black) represent the VIP score calculated for the differential metabolites in both groups. (**c**) Violin plots represent the *t*-test of the normalised concentrations of the significant metabolites (C10 = 0.1454 ± 0.02 mM (CI) vs. C10 = 0.2276 ± 0.04 mM (insomnia), *p*-value = 0.0027214, FDR = 0.032619; C8 = 0.0858 ± 0.01 mM (CI) vs. C8 = 0.132 ± 0.02 mM (insomnia), *p*-value = 0.002836, FDR = 0.032619; C14 = 0.0169 ± 0.002 mM (CI) vs. C14 = 0.0212 ± 0.001 mM (insomnia), *p*-value = 0.0036092, FDR = 0.032619; C12:1 = 0.1382 ± 0.005 mM (CI) vs. C12:1 = 0.1558 ± 0.006 mM (insomnia), *p*-value = 0.0036243, FDR = 0.032619; C18:1 = 0.0817 ± 0.009 mM (CI) vs. C18:1 = 0.1 ± 0.007 mM (insomnia), *p*-value = 0.0048163, FDR = 0.034677; and C18 = 0.0143 ± 0.001 mM (CI) vs. C18 = 0.0212 ± 0.003 mM (insomnia), *p*-value = 0.0072644, FDR = 0.043586). Dots (black) represent the metabolite concentration in each participant in the group, and the yellow dot in the centre represent the median of the corresponding metabolite concentration.

**Figure 3 metabolites-15-00417-f003:**
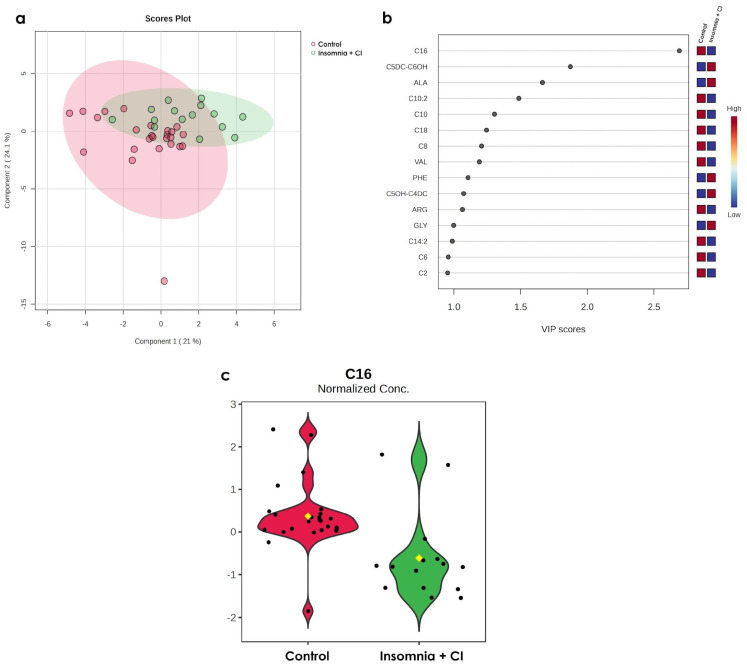
Comparison of the metabolomic profiles from the insomnia + CI and control groups. (**a**) PLS-DA analysis of the relative concentration in the serum metabolome in the control group (red dots represent the metabolome for each individual in the group) vs. the insomnia + CI group (green dots represent the metabolome for each individual in the group). The proportion of variance corresponds to Component 1: 21 and Component 2: 24.1%, Accuracy: 0.7, R2: 0.26785, Q2: −0.042916, and the permutation *p*-value = 0.0515. (**b**) VIP analysis represents the relative contribution of metabolites to the variance between groups. The colour scale (on the right side) represents metabolite concentration ranging from low (blue) to high (red). The VIP score cut-off has been adjusted to 1.0. Each dot (black) represent the VIP score calculated for the differential metabolites in both groups. (**c**) Violin plots represent the *t*-test of the normalised concentrations of the significant metabolite C16 (C16 = 0.0232 ± 0.0018 mM (control) vs. C16 = 0.01533 ± 0.0032 mM (insomnia + CI), *p*-value = 1.94 × 10^−4^, FDR = 0.007209). Dots (black) represent the metabolite concentration in each participant in the group, and the yellow dot in the centre represent the median of the corresponding metabolite concentration.

**Table 1 metabolites-15-00417-t001:** Demographics and clinical description of the study population.

Variables	Data
Age y.o. (mean ± SD)	70.6 ± 6.3
Sex	
Female (*n*, %)	57 (71.2)
BMI (kg/m^2^) (mean ± SD)	
Controls	27.93 ± 4.09
Cognitive impairment (CI)	27.40 ± 4.27
Insomnia	29.09 ± 3.81
Insomnia + CI	29.7 7± 4.47
Body composition	
Controls	
Fat mass (%) (mean ± SD)	33.03 ± 6.33
Weight (kg) (mean ± SD)	67.93 ± 11.38
CI	
Fat mass (%) (mean ± SD)	27.41 ± 4.27
Weight (kg) (mean ± SD)	64.18 ± 13.70
Insomnia	
Fat mass (%) (mean ± SD)	33.82 ± 7.79
Weight (kg) (mean ± SD)	71.00 ± 10.66
Insomnia + CI	
Fat mass (%) (mean ± SD)	36.6 ± 9.02
Weight (kg) (mean ± SD)	63.55 ± 11.41
Education in years (*n*, %)	
None (Illiterate)	3 (3.75)
<6	11 (13.75)
From 6 to 9	22 (27.5)
From 10 to 11	12 (15)
12 and more	32 (40)
Living arrangement	
Accompanied (*n*, %)	75 (93.75)
Multimorbidity (*n*, %)	20 (25)
Medication	
Number of medicines ± SD (*n*, %)	3.8 ± 2.9
Antidepressant consumption (*n*, %)	17 (21.3)
Anxiety (*n*, %)	17 (21.3)
Current smoking (*n*, %)	8 (10)
Current alcohol consumption (*n*, %)	14 (17.5)
Groups (*n*, %)	
Controls	22 (27.5)
CI	21 (26.3)
Insomnia	23 (46.3)
Insomnia + CI	14 (17.5)
Total of participants (*n*, %)	80 (100)

**Table 2 metabolites-15-00417-t002:** Acylcarnitine concentration ratios (mean ± SD).

	C16OH/C16	C16/C2	C18/C2
Control	0.6842 ± 0.1514	0.0255 ± 0.0115	0.0146 ± 0.0069
CI	0.7575 ± 0.2057	0.0263 ± 0.0128	0.0144 ± 0.0076
Insomnia	0.6550 ± 0.1318	0.0257 ± 0.0126	0.0144 ± 0.0088
Insomnia + CI	0.6727 ± 0.1429	0.0284 ± 0.0123	0.0159 ± 0.0067

## Data Availability

The data presented in this study are available on request from the corresponding author due to ethical restrictions. Further inquiries can be directed to the corresponding author.

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
