# Peer review of "Insomnia, Cognitive Impairment, or a Combination of Both, Alter Lipid Metabolism Due to Changes in Acylcarnitine Concentration in Older Persons"

_metabolites, 2025, doi:10.3390/metabo15060417_

Round 1

Reviewer 1 Report

Comments and Suggestions for Authors

In the manuscript titled “An exploratory analysis based on target metabolomics reveals that insomnia impairs lipid metabolism in older persons,” the authors characterize metabolomic profiles of insomnia and CI or in combination to address the pathways through which insomnia may lead to CI in the older population. 

1.    Dear authors, please could you re-write the title ensuring that it is more informative and appropriate. The title of the manuscript should have a clear, precise scientific meaning of the study.
2.    In the Materials and Methods you have subsection – “2.3. Insomnia classification” Actually you can’t classify insomnia with The Athens Insomnia Scale (AIS). 
3.    The study limitations should be mentioned in the special section or methods, but not in the conclusion section.

Author Response

Reviewer 1

In the manuscript titled “An exploratory analysis based on target metabolomics reveals that insomnia impairs lipid metabolism in older persons,” the authors characterize metabolomic profiles of insomnia and CI or in combination to address the pathways through which insomnia may lead to CI in the older population. 

Q1.    Dear authors, please could you re-write the title ensuring that it is more informative and appropriate. The title of the manuscript should have a clear, precise scientific meaning of the study.

Answer: In accordance with your valuable suggestion we have re-write the title of the manuscript. Now the title is: “Insomnia, cognitive impairment, or a combination of both, alter lipid metabolism due to changes in acylcarnitine concentration in older persons”.

Q2.    In the Materials and Methods you have subsection – “2.3. Insomnia classification” Actually you can’t classify insomnia with The Athens Insomnia Scale (AIS). 

Answer: We appreciate your valuable observation. Now, the subsection 2.3 appears as follows: 

“2.3. Insomnia evaluation symptoms

The Athens Insomnia Scale (AIS) is a wide-use self-report measure to evaluate insomnia symptoms and severity according to the International Classification of Diseases criteria version 10 (ICD-10) [19,20]. This test has been validated in Mexico, and consists of eight self-rating items. The first four items measure sleep problems quantitatively, the fifth item assesses sleep quality, and the remaining three evaluate the impact of insomnia during the day. The scores range from 0 to 24, with a score of ≥6 indicating the presence of insomnia [21]. The AIS Cronbach alpha value in this study was 0.90.”

Q3.    The study limitations should be mentioned in the special section or methods, but not in the conclusion section.

Answer: Thank you for your recommendation, accordingly we have mentioned the limitations in the Discussion section. Please refer to the main text for further details.

Reviewer 2 Report

Comments and Suggestions for Authors

The manuscript presents the results of studies of selected metabolites in the serum of elderly people with cognitive impairment (CI) and insomnia. The studies were performed using a targeted analysis based on MS, using a commercial method that allows for the determination of 40 metabolites, mainly amino acids and acylcarnitines. The authors focused on acylcarnitines, among which they found the most of changes. The subject of the work is interesting, but I have some doubts about the analysis and interpretation of the results.

The word acylcarnitine is not mentioned in the abstract, and symbols like C14, C18:1, etc. are often used to name fatty acids, so it may be confusing for the reader. Make it clear it in the abstract.

The aim of the study was to investigate how insomnia (I) can lead to CI, but the data were analyzed independently for pairs of groups: control vs CI, CI vs I, and control vs CI+I. In order to examine the relationship between I and CI, in my opinion, it would be necessary to perform the ANOVA statistical test for all the studied groups together.

The study participants were from the COSFORMA group, which included obese individuals. Obesity affects lipid metabolism. The paper lacks characteristics of the groups in the context of obesity, such as BMI. Differences in BMI in the groups could suggest a different than I and CI basis for changes in acylcarnitines . BMI and possibly other parameters related to obesity must be included in the table with the characteristics of the studied patient groups.

Does the decrease in acylcarnitines in the CI group indicate better mitochondrial function in this group than in the control group? Please refer to these changes in the discussion.

The discussion is focused on acylcarnitines. In the figures, the VIP score analysis also indicates differences in other metabolites, e.g. amino acids. Please refer to other metabolites besides acylcarnitines in the discussion.

In the conclusions, the authors again write about the relationship between CI and I, while as I wrote above, such conclusions cannot be drawn by analyzing CI vs control and CI vs I separately.

Author Response

Reviewer 2

The manuscript presents the results of studies of selected metabolites in the serum of elderly people with cognitive impairment (CI) and insomnia. The studies were performed using a targeted analysis based on MS, using a commercial method that allows for the determination of 40 metabolites, mainly amino acids and acylcarnitines. The authors focused on acylcarnitines, among which they found the most of changes. The subject of the work is interesting, but I have some doubts about the analysis and interpretation of the results.

Q1. The word acylcarnitine is not mentioned in the abstract, and symbols like C14, C18:1, etc. are often used to name fatty acids, so it may be confusing for the reader. Make it clear it in the abstract.

Answer: Thank you for your valuable commentary, the updated version of the manuscript contains such corrections in the abstract to avoid confusion among readers. 

Q2. The aim of the study was to investigate how insomnia (I) can lead to CI, but the data were analyzed independently for pairs of groups: control vs CI, CI vs I, and control vs CI+I. In order to examine the relationship between I and CI, in my opinion, it would be necessary to perform the ANOVA statistical test for all the studied groups together.

Answer: Thank you for your suggestion, the reviewed version includes the ANOVA test for the studied groups together. The results for such an analysis shows non-significant differences between groups when they are analyzed together (Accuracy= 0.28301, R2= 0.10064, and Q2= -0.16206). This data is included in Supplementary Information (Figure 1S), please refer to the main text for more details.

Q3. The study participants were from the COSFORMA group, which included obese individuals. Obesity affects lipid metabolism. The paper lacks characteristics of the groups in the context of obesity, such as BMI. Differences in BMI in the groups could suggest a different than I and CI basis for changes in acylcarnitines. BMI and possibly other parameters related to obesity must be included in the table with the characteristics of the studied patient groups.

Answer: Thank you for your valuable observation, accordingly the reviewed version of the manuscript include in Table 1 the following variables: BMI (kg/m2) (mean ± SD), Body composition (Fat mass (%) (mean ± SD), and Weight (kg) (mean ± SD) to support our findings. Additionally, we performed a statistical analysis, shown in Figure 1S from the Supplementary information, data showed non-significant differences between the BMI, Fat mass or weight among groups. These results strengthen our results about the effect of insomnia on CI on the metabolomic profiles. 

Q4. Does the decrease in acylcarnitines in the CI group indicate better mitochondrial function in this group than in the control group? Please refer to these changes in the discussion.

Answer: Thank you for your interesting question. In the reviewed version of the manuscript we address such a question by analyzing the AC/LC ratio. Particularly we analyzed the C16OH/C16, C16/C2 and C18/C2 ratios, which have been reported as representative measures of disturbed mitochondrial metabolism  (DOI: 10.1177/0883073895010002S03) which in turn relates to mitochondrial function (DOI: 10.3390/jcm10214855). In this sense, in section 3.3 we observed that the C16OH/C16 ratio is higher in all groups than the cut-off reported in previous studies (0.1-0.4) (DOI: 10.1007/s10545-012-9578-7), suggesting a potential mitochondrial dysfunction, particularly in the CI group that shows a slightly increase in the ratio as compared to the other groups. 

In the current version of the manuscript we include these results in the discussion, please refer to the main manuscript for further details. 

Q5. The discussion is focused on acylcarnitines. In the figures, the VIP score analysis also indicates differences in other metabolites, e.g. amino acids. Please refer to other metabolites besides acylcarnitines in the discussion.

Answer: Thank you for your observation. In metabolomics, a high VIP (Variable Importance in Projection) score indicates that a metabolite is important for separating groups. However, it doesn't necessarily mean that its concentration differs statistically between the groups. VIP scores prioritize metabolites that contribute most to the model's predictive power. Statistical tests (e.g., t-tests) are needed to determine if concentration differences are significant (DOI: 10.1080/07315724.2009.10719787). Therefore, since we did not observe significant differences in the concentration of AA, we avoid discussing these metabolites. In section 4. Discussion of the current version of the manuscript, we include this explanation to improve comprehension of our results. For more details, please refer to the main manuscript.

Q6. In the conclusions, the authors again write about the relationship between CI and I, while as I wrote above, such conclusions cannot be drawn by analyzing CI vs control and CI vs I separately.

Answer: Thank you for your valuable suggestion, in accordance, in the reviewed version of the manuscript we have updated our conclusion. For more details please refer to the main manuscript.

Round 2

Reviewer 2 Report

Comments and Suggestions for Authors

My comments have been addressed adequately.